# Newborn Screening for Pompe Disease: Pennsylvania Experience

**DOI:** 10.3390/ijns6040089

**Published:** 2020-11-13

**Authors:** Can Ficicioglu, Rebecca C. Ahrens-Nicklas, Joshua Barch, Sanmati R. Cuddapah, Brenda S. DiBoscio, James C. DiPerna, Patricia L. Gordon, Nadene Henderson, Caitlin Menello, Nicole Luongo, Damara Ortiz, Rui Xiao

**Affiliations:** 1Division of Human Genetics/Metabolism, Children’s Hospital of Philadelphia, Perelman School of Medicine at the University of Pennsylvania, Philadelphia, PA 19104, USA; ahrensNicklasR@email.chop.edu (R.C.A.-N.); cuddapahs@email.chop.edu (S.R.C.); dibosciob1@email.chop.edu (B.S.D.); menelloc@email.chop.edu (C.M.); Luongon@email.chop.edu (N.L.); 2Department of Pediatrics, Division of Medical Genetics, UPMC Children’s Hospital of Pittsburgh, Pittsburgh, PA 15224, USA; joshua.barch@chp.edu (J.B.); nadene.henderson@chp.edu (N.H.); damara.ortiz@chp.edu (D.O.); 3PerkinElmer, Mass Spectroscopy Unit, Pittsburgh, PA 15275, USA; James.DiPerna@perkinelmer.com; 4Division of Human Genetics, Penn State Heath Children’s Hospital, Penn State University College of Medicine, Hershey, PA 17033, USA; pgordon@pennstatehealth.psu.edu; 5Department of Pediatrics, Division of Biostatistics, Children’s Hospital of Philadelphia, Philadelphia, PA 19104, USA; xiaor@email.chop.edu; 6Department of Biostatistics, Epidemiology & Informatics, Perelman School of Medicine at the University of Pennsylvania, Philadelphia, PA 19104, USA

**Keywords:** Pompe disease, newborn screening, alpha glucosidase

## Abstract

Pennsylvania started newborn screening for Pompe disease in February 2016. Between February 2016 and December 2019, 531,139 newborns were screened. Alpha-Glucosidase (GAA) enzyme activity is measured by flow-injection tandem mass spectrometry (FIA/MS/MS) and full sequencing of the GAA gene is performed as a second-tier test in all newborns with low GAA enzyme activity [<2.10 micromole/L/h]. A total of 115 newborns had low GAA enzyme activity and abnormal genetic testing and were referred to metabolic centers. Two newborns were diagnosed with Infantile Onset Pompe Disease (IOPD), and 31 newborns were confirmed to have Late Onset Pompe Disease (LOPD). The incidence of IOPD + LOPD was 1:16,095. A total of 30 patients were compound heterozygous for one pathogenic and one variant of unknown significance (VUS) mutation or two VUS mutations and were defined as suspected LOPD. The incidence of IOPD + LOPD + suspected LOPD was 1: 8431 in PA. We also found 35 carriers, 15 pseudodeficiency carriers, and 2 false positive newborns.

## 1. Introduction

Pompe disease is an autosomal recessive inborn error of metabolism caused by mutations in the glucosidase alpha acid (*GAA*) gene located on the long arm of chromosome 17q25.2-q25.3. These genetic mutations cause deficient GAA enzyme activity [1]. Due to the deficiency of this enzyme, glycogen cannot be metabolized in lysosomes. This causes glycogen accumulation that damages cells throughout the body, especially muscle cells. Pathologic changes in muscle usually begin long before patients present with symptoms [2]. 

The previously estimated overall incidence of Pompe disease is 1 in 40,000 [1 in 138,000 for Infantile Onset Pompe Disease (IOPD) and 1 in 57,000 for Late Onset Pompe Disease (LOPD)] in the Netherlands [3]. This roughly corresponds to the incidence of clinically identified cases in New York state [4]. The frequency appears to vary significantly in different ethnic groups from 1 in 14,000 to 1 in 600,000 [1,3,4,5,6,7]. Recorded incidence is highest in African Americans and lowest in Portuguese populations. 

Patients with IOPD present with hypertrophic cardiomyopathy, hypotonia, macroglossia, feeding difficulties, and failure to thrive at around 2 months of age [8]. Some cases develop hypertrophic cardiomyopathy in utero. IOPD is rapidly progressive, and if left untreated, patients usually die in the first year of life. IOPD patients, diagnosed clinically or by newborn screening, always have elevated CK levels as a marker of muscle damage, and elevated urinary glucose tetrasaccharide (Glc4 or Hex4). Hex4 was shown to correlate with glycogen content in quadriceps biopsies in patients with IOPD [9]. Hex 4 is not only useful in the diagnosis of Pompe but also in monitoring the response to ERT [10].

Patients with LOPD present later in life with proximal muscle weakness, gait abnormalities, respiratory insufficiency, poor weight gain, and swallowing difficulties [11]. Individuals with LOPD do not typically develop hypertrophic cardiomyopathy, but some experience arrhythmias. LOPD patients who are diagnosed clinically often, but not always, have elevated CK and Hex 4 levels. Baseline evaluation of Hex 4 levels in LOPD cases detected through NBS have been reported to be within normal range [10]. 

*Diagnostic Odyssey*: Before implementation of newborn screening, there was, on average, a 3 month- delay in diagnosing IOPD after the onset of symptoms [12]. In LOPD, symptoms may begin anytime from infancy to adulthood. In pediatric onset cases, symptom onset occurs at an average of 6 years of age, yet the diagnosis is made at an average of 18 years of age. Therefore, on average, there is a 12-year delay in diagnosis [12]. The average age of symptom onset in adult-onset LOPD is 35 years of age, with a 7-year delay in diagnosis after symptom onset [12]. Non-specific symptoms, seen in other more common disorders, coupled with lack of knowledge about this rare disorder usually result in significant diagnostic delays. These factors importantly shape an individual patient’s odyssey and outcomes with all forms of Pompe disease.

Enzyme replacement therapy (ERT) has been available since 2006 for all forms of Pompe disease. It has dramatically changed patient outcomes. [13]. ERT improves cardiac and skeletal symptoms and slows down the progression of disease. Patients with severe disease, who cannot produce any natural GAA enzyme, are classified as Cross-Reactive Immunological Material (CRIM) negative. In such CRIM negative patients, the immune system identifies the administered enzyme as foreign and produces antibodies that make the enzyme therapy less effective [14]. Immune modulation therapy can be used to inhibit the development of antibody response in CRIM negative patients at the time of the start of their ERT [15,16,17]. 

This life-changing therapy is more effective when initiated before the onset of symptoms [18,19,20,21,22]. When we consider that (a) pathology begins long before patients present with symptoms and (b) many patients have a protracted diagnostic odyssey, shortening the diagnostic delay through newborn screening and beginning treatment as soon as possible is crucial for all forms of Pompe disease. 

A pilot newborn screening (NBS) was launched in Taiwan in 2005. Chien et al. demonstrated the importance of NBS not only for IOPD but also for LOPD [19,20,21]. Between 2005 and 2009, 344,056 newborns were screened in Taiwan, and 13 cases of LOPD were detected [21]. A total of 4 of 13 patients were put on ERT because of hypotonia, muscle weakness, delayed developmental milestones/motor skills, or elevated CK levels starting at the ages of 1.5, 14, 34, and 36 months. Muscle biopsy specimens obtained from the treated patients revealed increased storage of glycogen and lipids [21]. Soon after Taiwan, several pilot programs in Italy, Australia, Japan, Korea, USA, and Hungary tested the feasibility of Pompe NBS and reported the incidence of and the impact of the disease [23]. 

In the USA, Pompe disease was added to the recommended universal newborn screening panel (RUSP) in February 2015 [24]. Some states such as Missouri, Illinois, and New York started screening for Pompe before the RUSP recommendation. Pennsylvania started Pompe screening in February 2016. In this paper, we present Pennsylvania data and experience on Pompe NBS. Our aim is three-fold: to augment state-based knowledge of the disease and its diagnosis through NBS; to document the benefits and challenges of NBS for Pompe disease in Pennsylvania; and to encourage the speedy adoption of NBS for Pompe by other states and countries. 

## 2. Materials and Methods

In Pennsylvania, NBS testing is done by PerkinElmer newborn screening laboratory in Pittsburgh, PA. GAA enzyme activity is measured by flow-injection tandem mass spectrometry (FIA/MS/MS) using the algorithm shown in Figure 1 [25]. Blood is obtained at around 36 h of life. A second sample is requested if the first GAA enzyme activity is below the cutoff value. If this second GAA enzyme activity is below the cutoff, the second-tier test, *GAA* full gene sequencing (Next Generation Sequencing), is performed. In Pennsylvania, newborn screening is free, and both GAA enzyme level and full gene sequencing are done as part of Pompe NBS without any charges. The cutoff was established at ~18% of the apparently normal newborn mean GAA activity and is based on the analysis of ~1000 apparently normal newborns as well as 9 confirmed positive Pompe patients. In PerkinElmer laboratory, it was determined that the daily patient means are stable, and thus, the use of a fixed cutoff has been utilized and has been shown to be effective. The use of fixed cutoffs also makes the daily evaluation of data much less complex. Results falling below this cutoff are considered abnormal.

A single 1/8-inch (3mm) Dried Blood Spot (DBS) disc is extracted with 30 µL of cocktail solution containing β-Glucocerebrosidase (ABG), Acid Sphingomyelinase (ASM), Alpha-Glucosidase (GAA), Alpha-Galactosidase (GLA), Galactocerebrosidase (GALC) and Alpha-Iduronidase (IDUA) substrates and internal standards (in a buffer of pH 4.8) at 37 °C for 18 h +/− 2 h. The enzymatic reactions are quenched with 50% ethyl acetate: methanol followed by a liquid–liquid extraction using 50% ethyl acetate: water.

An aliquot of the organic top layer is transferred into a clean deep well plate, dried under a gentle stream of nitrogen gas, reconstituted in 80%/20%/0.2% acetonitrile/water/formic acid solution, and subjected to flow-injection tandem mass spectrometry (FIA/MS/MS). The electrospray source is operated in positive mode, and the analytes are interrogated in multiple reaction monitoring (MRM) mode. Blank filter paper spots are analyzed for background correction.

Enzyme activities (in units of µmol/L/h) are defined as the amount of substrate hydrolyzed by the enzyme in the reaction and are determined by calculating the ion abundance ratio of product to internal standard, multiplied by the volume and concentration of internal standard, divided by the response factor ratio of product to internal standard, sample incubation time, and sample blood volume (Enzyme activity = (P/IS)*[IS]*V_IS_/RF/3.1 µL/time). A sample volume of 3.1 µL is assigned to a 1/8-inch DBS punch. A fixed GAA enzyme activity cutoff of 2.10 micromole/L/h is utilized. A repeat specimen is requested on an initial GAA enzyme result of <2.10 µmol/L/h. A repeat specimen with a second GAA enzyme level of <2.10 µmol/L/h is reflexed to full *GAA* gene sequencing. Turn-around-time for GAA enzyme testing is <72 h from specimen receipt. Full *GAA* sequencing turn-around time is 7–10 days from reflex.

Newborns with low enzyme activity (<2.10 µmol/L/h) and at least one variant [pathogenic, likely pathogenic, or variant of unknown significance (VUS)] are referred to one of the seven metabolic referral centers in Pennsylvania by pediatricians and department of health newborn screening program. Newborns with low enzyme activity and sequencing revealing only pseudodeficiency allele(s) are not referred to metabolic centers by the state. However, some pediatricians refer them to the metabolic centers for further clarification.

After the initial metabolic evaluation and confirmatory test results, patients are classified as IOPD, LOPD, “suspected” LOPD, or carrier. If patients carry one pathogenic variant and one VUS or two VUS, they are defined as “suspected” LOPD. After genetic counseling, carriers and those with pseudo-deficiency mutations are discharged with no further follow-up; all others are followed by a metabolic geneticist. Although there are slight differences among Pennsylvania centers in the initial work up, confirmatory GAA enzyme activity, aspartate aminotransferase (AST), alanine aminotransferase (ALT), creatine kinase (CK), urine glucose tetrasaccharide (Glc4 or Hex4) levels are tested in all cases. At the outset of NBS for Pompe, all centers performed electrocardiogram (ECG) and echocardiogram (ECHO) at the time of referral. As centers gained more experience, some decided not to perform ECG and ECHO at the first visit in cases with late onset mutations. Patients with LOPD and suspected LOPD are usually seen every 3 months in the first year and subsequently, every 6 months. In IOPD patients, treatment, including appropriate immunomodulation, is initiated immediately.

Statistical Analysis: The demographic and laboratory data were summarized by median and interquartile range (IQR) for continuous variables and frequency and percentage for categorical variables. Wilcoxon rank sum test was used to compare the GAA enzyme activity, as well as AST, ALT, CK, Hex4 levels between groups, either disease groups or genotype groups. Two-sided *p* values less than 0.05 were considered statistically significant. All data analyses were performed using SAS 9.4. 

The Institutional Review Board reviewed and deemed this study exempt (IRB 20-017914). 

## 3. Results

Between February 2016 and December 2019, 531,139 newborns were screened for Pompe disease. A total of 180 newborns (0.03%) had a low GAA enzyme activity on the first NBS. Repeat GAA enzyme activity from a second filter paper specimen was lower than the cutoff in 115 newborns (0.02%), and subsequently full *GAA* gene sequencing was performed. The screening algorithm is shown in Figure 1. Final diagnosis, screening and biochemical data from patients with a confirmed diagnosis are summarized in Table 1, Table 2 and Table 3. All positive cases were referred to one of the seven metabolic centers. The incidence of IOPD + LOPD was 1:16,095 and IOPD + LOPD + suspected LOPD was 1:8431 in PA. Based on the PA DOH records, 5 newborns with positive screening were lost to follow up and never seen in metabolic centers. All 5 newborns were carriers based on the genetic test result.

IOPD: Two patients were diagnosed with IOPD. Their GAA enzyme levels were both 0.65 micromole/L/h at the first screening and 0.22 and 0.34 at the second screening, respectively. They both had lower GAA enzyme activities compared to carriers and pseudodeficiency; yet, there was overlap with other forms of Pompe disease. Both patients were compound heterozygous for mutations previously reported in patients with IOPD (Table 2).

The first patient was in the hospital and admitted for hypertrophic cardiomyopathy when the NBS was reported. He had elevated CK, AST, ALT, BNP, and Hex4 levels. ECG showed biventricular hypertrophy. ECHO findings included (1) moderately to severely decreased left ventricular systolic wall motion, (2) moderate hypertrophy of the left ventricle, and (3) severe right ventricular hypertrophy. The CRIM status was determined based on the genotype. ERT with immunomodulation (CRIM negative protocol) was started at 21 days of age [16]. At 31 months of age, he has normal cardiac functions but global developmental delay. He has obstructive sleep apnea and requires BiPAP at night. He continues to receive physical, occupational, and speech therapies. 

The second patient had elevated CK, AST, ALT, BNP, and Hex4 levels. ECG showed short PR interval and biventricular hypertrophy. ECHO findings were (1) severe septal left ventricular hypertrophy and (2) mildly hypertrophied right ventricle. He was CRIM positive based on his genotype. ERT with immuno-modulation (methotrexate only) was started at 10 days of life [17]. His cardiac functions stabilized by 4 weeks on this regimen and normalized within 3 months. His CK never rose above 900, and down trended after the 3rd infusion and has been normal since then. He had mild motor delay on exam at 6 months old.

LOPD: Thirty-one newborns were identified to have LOPD (Table 2). Twelve of them (39%) were homozygous for the most common splice site mutation (c.-32-13T>G). A total of 16 patients were compound heterozygous, and 11 of them carried the splice site mutation (c.-32-13T>G). Three cases were homozygous for c.2238G>C mutation.

All patients had low confirmatory GAA enzyme activity except two patients who had levels within normal range (patient 5, and patient 16 in Table 2). Both of these patients were homozygous for the c.-32-13T>G mutation. CK levels ranged from 48 to 617 U/L, with 11 individuals demonstrating mildly elevated CK levels. Urine Hex4 levels were within normal range in all cases. All but 2 cases had ECGs at the initial visit. ECHO was done at the time of initial visit in 24 patients. Two cases had their first ECHO at 3 months and 6 months, respectively. One case showed mild ventricular septal hypertrophy and mild hypertrophy of the left ventricle at the time of confirmatory test but follow up ECG and ECHO were normal at 3 months of age. Otherwise, ECG and ECHO at the time of diagnosis were normal or demonstrated nonspecific findings. Chest radiograms were done only by one referral site, and they were normal. 

Suspected LOPD: Thirty patients were compound heterozygous for one pathogenic variant and one variant of unknown significance (VUS) or two VUS and were defined as suspected LOPD (Table 3). A total of 16 of 30 carried the most common splice site (c.-32-13T>G) mutation. The confirmatory GAA enzyme activity was lower than normal in most cases; three individuals had levels within the low-end of the normal range (Table 3). CK levels ranged from 65 to 668 U/L. Six cases had slightly elevated CK levels at the time of initial visit. Urine Hex4 levels were normal in all suspected LOPD patients. ECGs were normal or showed non-specific findings. A total of 20 cases had ECHO at the time of diagnosis, and they were normal demonstrated non-specific findings such as PFO or secundum ASD.

Pseudodeficiency: Fifteen infants had one or two pseudodeficiency alles, and they were reported as pseudodeficiency carriers. Although a referral to a metabolic center was not made, PCPs were given an option to consult with a metabolic physician about these cases. A total of 8 of 15 newborns with pseudodeficiency were referred to metabolic centers to enhance families’ understanding of the meaning of pseudodeficiency alleles. 

Carriers: Thirty-five newborns had low enzyme activity and one mutation. Further work up, including deletion and duplication testing, confirmed that they were carriers for Pompe disease. Parents of these infants received genetic counseling in each case. 

False positive: There were two newborns who had GAA enzyme activities 1.78 and 1.61 (>2.0 micromole/L/h) on the second NBS, respectively, but they had negative *GAA* gene sequencing. They were seen at a metabolism referral center, and their confirmatory enzyme levels were 4,6 (>3.88) and 5.6 (>3.88), respectively. Both were reported as false positive.


**Questions raised by Screening Results**


Biomarker data collected through NBS for Pompe disease raised several important questions.


**Do NBS GAA and confirmatory GAA levels make it possible to distinguish different types of Pompe disease, pseudodeficiencies, and carriers?**


The median level of GAA in LOPD patients was lower than those of suspected LOPD, carriers, and pseudodeficiency cases. Patients with LOPD had significantly lower levels of GAA enzyme activity compared to cases who are carriers or have pseudodeficiency alleles in all NBS and confirmatory tests (*p* < 0.003, 0 and 0.007). There was no statistically significant difference in GAA levels between LOPD and suspected LOPD cases except the second NBS (*p* < 0.015). Patients with suspected LOPD also had statistically lower GAA levels in all NBS and confirmatory tests compared to those of carriers/pseudodeficiency (*p* < 0.029, 0.002, 0.001) (Figure 2). 

*Answer*: In our cohort, the median value of GAA enzyme levels were statistically different, and it was possible to differentiate LOPD and suspected LOPD cases from pseudodeficiency and carriers. Two IOPD cases had much lower NBS GAA enzyme values compared to others. Although there were statistically significant differences, it is not always possible to utilize GAA levels to identify the Pompe disease status in individual newborns because of overlapping GAA values. 


**Do initial AST, ALT, CK, and Hex4 levels make it possible to distinguish different types of Pompe disease?**


The median values of AST (72.5 U/L) and ALT (42.5 U/L) were higher in LOPD patients compared to those of suspected LOPD (AST: 64 U/L, ALT: 36 U/L) and carriers and pseudodeficiency (AST: 63.5 U/L, ALT: 32 U/L). The statistical difference was only noted in ALT values between LOPD and suspected LOPD (*p* < 0.048), and LOPD and carriers/pseudodeficiency (*p* < 0.002) (Figure 3).

The median value of CK (265 U/L) was significantly higher in LOPD patients compared to the median values of the suspected LOPD (123 U/L) or carriers/pseudodeficiency (107) (*p* < 0.034 and 0, respectively) (Figure 4).

Although the median value of Hex 4 (5.23 mmol/mol creatinine) was higher in LOPD, it was not statistically different than those of suspected LOPD (4.5 mmol/mol creatinine) or carrier/pseudodeficiency (3.95 mmol/mol creatinine) (Figure 4). 

None of these biomarkers were significantly different between suspected LOPD and carriers/pseudodeficiency (Figure 3 and Figure 4).

*Answer*: In our cohort, the median values of CK and ALT (but not Hex4 or AST) levels were significantly higher in LOPD compared to other forms of Pompe disease detected by NBS. Two IOPD cases had much higher CK, AST, ALT and Hex4 levels compare to others. It is important to note that the analytes alone may not be used to make phenotyping or clinical decision in all cases.


**Do cases homozygous for the common splice site mutation (c.-32-13T>G) have different GAA enzyme or CK levels compared to other LOPD patients?**


The median values of GAA enzyme activity (NBS#2: 0.81 µmol/L/h, Confirmatory GAA: 3.3) were higher, and the median value of CK level (204 U/L) was lower in the patients homozygous for c.-32-13T>G mutation than those of other LOPD patients (NBS#2: 0.56 µmol/L/h, confirmatory GAA:2.2, CK:324 U/L), but we could not find any statistically significant difference (Figure 5). 

*Answer*: No, GAA enzyme or CK levels do not help to differentiate cases homozygous for common splice site mutation from LOPD cases caused by other pathogenic mutations. 

## 4. Discussion

Newborn screening detects all subtypes of Pompe disease. The reported incidence of the disease has increased since initiation of NBS, with values ranging from 1:8684 to 1:23,596 [23,26,27]. Increased incidence is largely due to improved detection of LOPD and suspected LOPD cases through NBS. In Pennsylvania, we found a similar increase with a combined incidence of IOPD and LOPD of 1:16,095 and a combined incidence of IOPD, LOPD, and suspected LOPD of 1:8,431. The incidence of IOPD by itself was 1:265,570 in Pennsylvania.

NBS for Pompe is done either by measuring only GAA enzyme activity or both GAA enzyme activity and full gene sequencing. In Pennsylvania, we chose to perform both GAA enzyme activity and gene sequencing in the NBS to increase certainty of diagnosis and provide a more detailed definition of positive screening at the initial patient visit [28]. Performing genetic testing as part of NBS has multiple advantages. The genetic test identifies those with pseudodeficiency alleles that are common in Pompe. Such patients do not need referral to a metabolic center for additional workup. Having a molecular diagnosis at the time of first visit informs the discussion of the different forms of the disease. It decreases anxiety by offering families a final diagnosis and a follow up plan based on the NBS test result. The availability of genotype at the time of reporting of abnormal NBS can also help in predicting CRIM status in IOPD patients and facilitates an informed and swift decision about the need for immunomodulation. In the absence of the results of genotype testing, families are subjected to uncertainty of diagnosis during the minimum two-week period required to order and receive the results of the genetic test. Including genetic testing with NBS for Pompe also circumvents the risk that insurance companies might deny coverage for it and guarantees that patients do not bear any additional financial burden. 

NBS detects both IOPD and LOPD. Detecting IOPD cases in the first weeks of life is essential for initiation of therapy to yield optimal outcomes [21,29]. In our cohort, 2 patients with IOPD were detected (the incidence of IOPD by itself was 1:265,570 in Pennsylvania) and treated with ERT. The first patient was in the NICU of a local hospital and evaluated for feeding difficulty and cardiomyopathy. The physicians started a diagnostic work up to rule out potential etiologies and waited for the final result of Pompe NBS, which showed that he had IOPD at 19 days of age. This case showed us that there is a risk of delaying diagnosis of IOPD in the current Pompe algorithm in Pennsylvania. A second specimen is necessary to complete the sequencing test due to depletion of blood on the initial specimen on which all other newborn screening tests are performed. After detecting the first case of IOPD, we updated the NBS algorithm in Pennsylvania for Pompe to minimize delays in diagnosis of IOPD. The updated algorithm requires department of health (DOH) nurses to ask pediatricians if newborn has cardiomyopathy or symptoms of Pompe disease such as hypotonia and feeding issues when the first abnormal test is reported. 

We identified 31 patients with LOPD. In addition to molecular testing results, biochemical findings such as GAA enzyme activity and CK level may help to differentiate LOPD cases from suspected LOPD or carriers/pseudodeficiency at the initial visit. In general, in LOPD cases as compared to suspected LOPD and carriers/pseudodeficiency carriers, GAA enzyme activities were lower, and CK levels were higher. Hex 4 levels were elevated and helpful biomarker for IOPD cases, but it was normal in LOPD cases at the time of diagnosis. 

Since LOPD patients may present with symptoms at any age from infancy to adulthood, some may question the benefits and consequences of detection of LOPD patients via NBS. Early detection may increase family anxiety [30]. However, genetic counseling and clear explanations about LOPD along with a detailed follow-up plan reduce family anxiety accompanying initial diagnosis. We know that an extended delay in diagnosis occurs in most LOPD patients, which adversely impacts outcomes. Outcomes of early treatment of LOPD patients detected by NBS in Taiwan have been promising (21). Several metabolic centers in Pennsylvania closely follow patients with LOPD detected by NBS to make judgments about when to start treatment based on very early signs and symptoms of disease (biomarkers and developmental tests) in each individual patient. 

The leaky *GAA* splice site variant, c.-32-13T>G in intron 1 is found on at least one allele in 68–90% of Caucasian patients [31]. In our NBS cohort, 74% had at least one c.-32-13T>G variant. There was no statistically meaningful difference in GAA enzyme or CK levels at the time of diagnosis between LOPD patients homozygous for c.-32-13T>G and other pathogenic mutations. Patients homozygous for c.-32-13T>G usually develop mild LOPD. However, recent reports described patients with more classical LOPD and highlighted the risk for arrhythmias in these patients [32,33,34]. These reports emphasize the importance of close follow up and detailed evaluations in all Pompe patients regardless of genotype.

We also identified 30 patients with suspected LOPD, i.e., they harbor two VUS or one pathogenic allele and one VUS. In this group, average GAA enzyme activity was significantly lower than those of carriers and cases with pseudodeficiency alleles on both NBS and confirmatory tests. The values of biomarkers such as CK, Hex4, AST, and ALT did not differ from those seen in carriers or pseudodeficiency carriers. This group poses a significant challenge. Some of these cases might never develop symptoms of Pompe disease; however, given our inability to fully assess disease risk in this cohort, they require ongoing monitoring. We follow up these patients every 6 months. 

Pennsylvania data differ from those of Illinois and Missouri. In Illinois, a total of 684,290 infants were screened between 3 November 2014 and 30 September 2019 [27]. A total of 395 newborns with positive NBS were referred to metabolic centers. A total of 234 of 395 (59%) were false positive (normal confirmatory enzyme activity), and 62 (26%) carried pseudodeficiency alleles; this compares to only 1.7% and 13% in Pennsylvania, respectively. In Missouri, 467,000 newborns were screened for Pompe disease, and 274 had a positive test based on decreased GAA activity between January 2013 and December 2018 [35]. A total of 97 of 274 (35%) were false positive based on normal GAA activity, and 53 of 274 (19%) carried pseudodeficiency alleles; this compares to only 1.7% and 13% in Pennsylvania, respectively. These differences may arise from the fact that (1) Pennsylvania measures GAA activity twice on two different samples; Illinois and Missouri only test once; (2) each state makes its own decision about the chosen cutoff value for GAA; and (3) each state has different percentages of ethnic groups in the population. 

## 5. Conclusions

The Pennsylvania experience shows that the overall incidence of Pompe cases increased after initiation of NBS compared to previously reported incidence [3,4,5,6,7] due to increased numbers of LOPD cases detected by screening. Furthermore, in our experience, NBS can detect suspected LOPD cases, some of whom may never go on to develop symptoms. Genotyping as a second-tier test was essential to inform the final diagnosis. This timely molecular diagnosis facilitated clear results disclosure at the first clinical visit and reduced parental anxiety. Finally, false positive and pseudodeficiency cases occurred at much lower rates than previously reported in other states. 

Close monitoring and data collection on all patients detected through NBS is essential to assess the long-term outcomes and success of NBS for Pompe disease. NBS registries should be created and funded to enable data collection. Increased data collection will make it possible to identify and understand the pathogenic vs benign nature of VUS in cases defined as suspected LOPD. 

## Figures and Tables

**Figure 1 IJNS-06-00089-f001:**
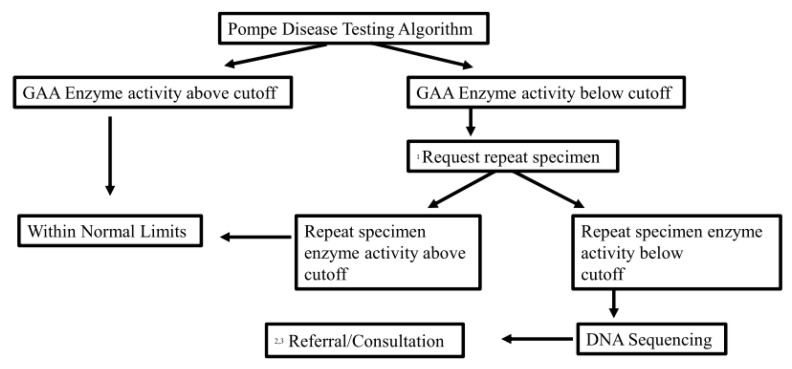
Flow chart of screening algorithm. ^1^ Pompe Repeat Request: Results show decreased enzyme activity for lysosomal alpha-glucosidase (GAA). We recommend a repeat dried filter paper blood specimen within 48 hours after the reporting of the first abnormal test. You should discuss the case with one of the metabolism referral centers if newborn has cardiomyopathy or symptoms of Pompe disease such as hypotonia and feeding issues. ^2^ Pompe Referral (Sequencing positive): Results continue to show decreased enzyme activity for Lysosomal alpha-Glucosidase (GAA). This result may be associated with Pompe disease. We recommend referral to a metabolic specialist. ^3^ Pompe Consult (Sequencing pseudodeficiency or no variant): Results continue to show decreased enzyme activity for Lysosomal alpha-Glucosidase (GAA). Consultation with a metabolic specialist may be considered to review and interpret these results in context with the patient’s clinical presentation.

**Figure 2 IJNS-06-00089-f002:**
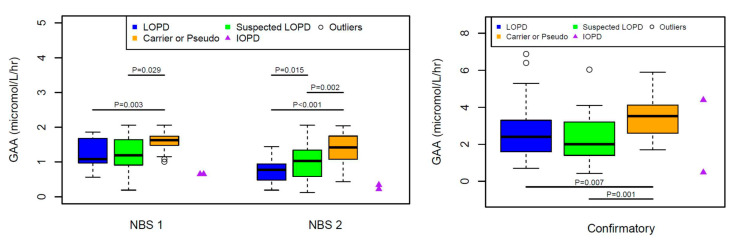
Comparing NBS and Confirmatory GAA enzyme activities of LOPD to Carriers/Pseudodeficiency or to Suspected LOPD patients, and suspected LOPD to carriers/pseudodeficiency) using Wilcoxon rank sum test. **NBS GAA#1** (Median: Min, Max): LOPD (*n*:22) (1.08:0.56,1.86); Suspected LOPD (*n*:21) (1.19: 0.19, 2.06); Carriers or Pseudo (*n*:30) (1.62: 1.01, 2.06), **NBS GAA#2** (Median: Min, Max): LOPD (*n*:30) (0.77:0.19, 1.44); Suspected LOPD (*n*:30) (1.03:0.12, 2.06); Carriers or Pseudo (*n*:43) (1.42:0.43, 2.04), **Confirmatory GAA** (Median, Min, Max): LOPD (*n*:23) (2.4:0.7, 6.9); Suspected LOPD (*n*:23) (2: 0.43, 6.05); Carriers or Pseudo (*n*:28) (3.5:1.7, 12.4).

**Figure 3 IJNS-06-00089-f003:**
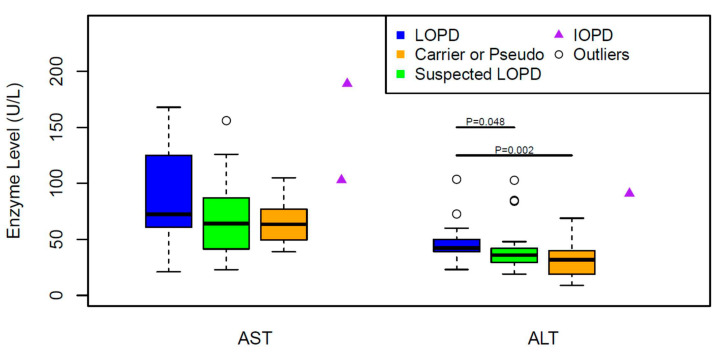
Comparing AST and ALT values of LOPD to Carriers/Pseudodeficiency or to Suspected LOPD patients, and suspected LOPD to carriers/Pseudodeficiency using Wilcoxon rank sum test. **AST** (U/L) (Median:Min, Max): LOPD (n:22) (72.5:21.1,168); Suspected LOPD (*n*:24) (64:23, 156); Carriers or Pseudo (*n*:20) (63.5:39, 105), **ALT**(U/L): LOPD (*n*:22) (42.5:23.1, 104); Suspected LOPD (*n*:24) (36:19, 103); Carriers or Pseudo (*n*:22) (32:9, 69).

**Figure 4 IJNS-06-00089-f004:**
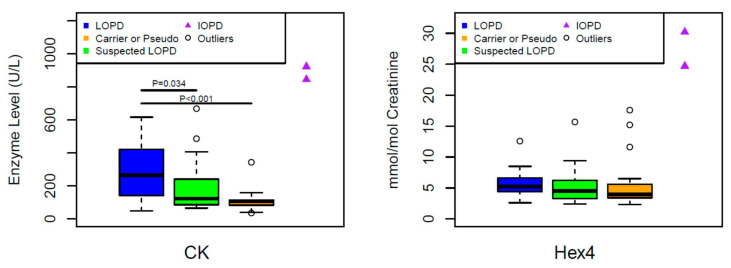
Comparing CK and Hex4 levels of LOPD to Carriers/Pseudodeficiency or to Suspected LOPD patients, and suspected LOPD to Carriers/Pseudodeficiency using Wilcoxon rank sum test. **CK** (U/L) (Median:Min, Max): LOPD (*n*:25) (265:48, 617); Suspected LOPD (*n*:25) (123:65, 668); Carriers or Pseudo (*n*:27) (107:35, 344), **Hex4** (mmol/mol creatinine): LOPD (*n*:21) (5.23:2.6, 12.6); Suspected LOPD (*n*:24) (4.5:2.4, 15.7); Carriers or Pseudo (*n*:26) (3.95:2.3, 17.6).

**Figure 5 IJNS-06-00089-f005:**
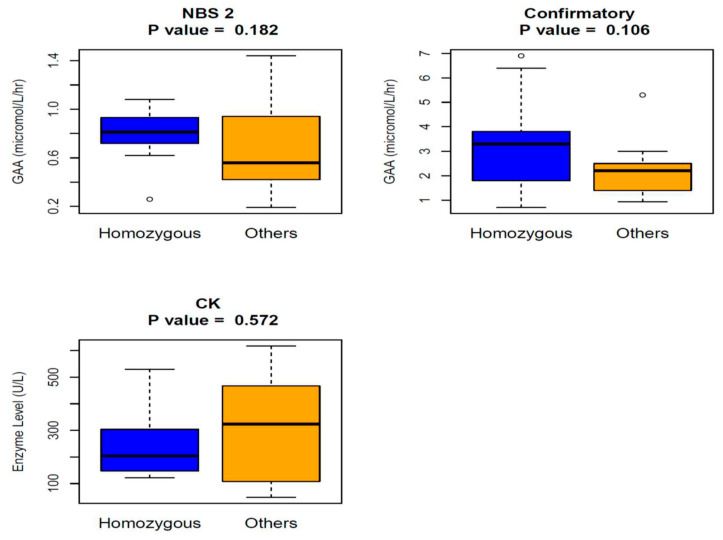
Comparison of NBS #2, Confirmatory GAA enzyme activities, and CK between c.-32-13T>G mutation homozygous patients and others in LOPD group, using Wilcoxon rank sum test. **NBS GAA#2** (Median:Min, Max): LOPD (homozygous for c.-32-13T>G) (*n*:12) (0.81:0.26, 1.08); LOPD-other pathogenic mutations (*n*:18) (0.56:0.19, 1.44); **Confirmatory GAA**: LOPD (homozygous for c.-32-13T>G) (*n*:10) (3.3:0.7, 6.9); LOPD-other pathogenic mutations (*n*:13) (2.2:0.93, 5.3); CK (U/L): LOPD (homozygous for c.-32-13T>G) (*n*:11) (204:122, 530); LOPD-other pathogenic mutations (*n*:14) (324:48, 617).

**Table 1 IJNS-06-00089-t001:** Final diagnosis of newborn screening for Pompe disease.

Pompe Disease	*n*	Incidence
IOPD	2	1:265,570
LOPD	31	1:17,134
Suspected LOPD	30	1:17,705
IOPD+LOPD	33	1:16,095
IOPD+ LOPD+ suspected LOPD	63	1:8431
Carriers	35	1:15,175
Pseudodeficiency	15	1:35,409
False positive (Normal confirmatory GAA activity, negative GAA full gene sequencing)	2	1:265,570

Total newborns screened: 531,139, Total positive *n* (%):115 (0.02).

**Table 2 IJNS-06-00089-t002:** Biochemical Parameters of Screened Patients confirmed IOPD and LOPD.

	Newborn Screening			Confirmatory Testing				
	GAA#1 (>2.10 micromole/L/h)	GAA#2 (>2.10 micromole/L/h)	Genotype	GAA	AST and ALT (Normal Range for Lab, U/L)	CK (Normal Range for Lab, U/L)	BNP (Normal Range for Lab, pg/mL)	Hex4 (Normal Range for Lab, (mmol/mol Creatinine)
**IOPD Patients**								
1	0.65	0.22	**c.759delC/c.1551+1G>C**	4.4 (>6.7)	AST: 103 (30–100)	923 (60–305)	1080 (0.0–100.0)	24.7
2	0.65	0.34	**c.525delT/c.1694_1697delTCTC**	0.4 (<3.88)	AST: 189 (22–71); ALT: 91 (7–50)	846 (60–305)	1232.5 (0.0–100.0)	30.2 (<20)
**LOPD Patients**								
1	1.86	0.78	**c.-32-13T>G/c.-32-13T>G**	3.1 (>6.7)	AST: 72 (24–72); ALT: 41 (17–63)	530 (28–300)	ND	4.9 (≤20)
2	1.86	1.44	**c.2238G>C/c.1552-3C>G**	5.3 (>6.7)	AST: 21 (24–72); ALT: 23 (17–63)	72 (28–300)	ND	4.2 (≤20)
3	1.73	1.08	**c.-32-13T>G/c.-32-13T>G**	3.6 (>6.7)	AST: 56 (24–72); ALT: 49 (17–63)	125 (28–300)	ND	4.5 (≤20)
4	1.24	0.67	**c.-32-13T>G/c.-32-13T>G**	2.4 (>6.7)	AST: 66 (24–72); ALT: 52 (17–63)	361 (28–300)	ND	3.9 (≤20)
5	1.82	0.94	**c.-32-13T>G/c.-32-13T>G**	6.9 (>6.7)	AST: 52 (24–72); ALT: 46 (17–63)	333 (28–300)	ND	4.4 (≤20)
6	1.32	0.88	**c.-32-13T>G/c.-32-13T>G**	0.7 (>6.7)	AST: 61 (24–72); ALT: 39 (17–63)	176 (28–300)	ND	5.9 (≤20)
7	1.05	0.19	**c.-32-13 T>C/c.546 G>A**	2.2 (>3.88)		283 (30–135)	ND	2.6 (<3.0)
8	0.83	0.85	**c.-32-13 T>G/c. 1655 T>C**	0.93 (>3.88)		365 (60–305)	ND	5.23 (<−8.9)
9	0.97	0.33	**c.2238G>C/c.2281delGinsAT/** *c.2065G>A*	0.93 (>3.88)	AST: 72 (22–71); ALT: 43 (7–50)	145 (60–305)	13.0 (0.0–100.0)	5.3 (≤20)
10	0.84	0.49	**c.1438-1G>C/c.-32-13T>G**	2.1 (>3.88)	AST: 168 (20–64); ALT: 104 (12–42)	617 (60–305)	-	6.6 (≤20)
11	-	0.63	**c.-32-13T>G/c.2560C>T**	3.00 (>3.88)	AST: 125 (22–71); ALT: 50 (7–50)	421 (60–305)	-	-
12	-	-	**c.-32-13T>G /c.2238G>C**	2.60 (>3.88)	AST: 112 (22–71); ALT: 50 (7–50)	464 (60–305)	51.6 (0.0–100.0)	6.1 (≤20)
13	0.56	0.62	**c.-32-13T>G/c.-32-13T>G**	3.80 (>3.88)	AST: 78 (20–64); ALT: 42 (12–42)	122 (60–305)	71.8 (0.0–100.0)	4.8 (≤20)
14	1.68	0.79	**c.156_157delTC/c.-32-13T>G**	2.40 (>3.88)	AST: 166 (20–64); ALT: 73 (12–42)	467 (60–305)	-	7.0 (≤20)
15	1.11	0.5	**c.2238G>C/c.-32-13T>G/** *c.2065G>A*	1.60 (>3.88)	AST: 49 (20–64); ALT: 27 (12–42)	68 (60–305)	-	2.8 (≤20)
16	1.06	0.26	**c.-32-13T>G/c.-32-13T>G**	6.4 (>3.88)	AST: 73 (22–71); ALT: 39 (7–50)	140 (60–305)	99.8 (0.0–100.0)	8.5 (≤20)
17	1.04	0.77	**c.-32-13T>G/c.-32-13T>G**	1.80 (>3.88)	AST: 150 (22–71); ALT: 42 (7–50)	275 (60–305)	44.5 (0.0–100.0)	5.2 (≤20)
18	0.7	0.34	**c.-32-13T>G/c.2560C>T**	1.40 (>3.88)	AST: 152 (22–71); ALT: 55 (7–50)	542 (60–305)	16.5 (0.0–100.0)	7.6 (≤20)
19	1.07	0.38	**c.456_458insGA/c.-32-13T>G**	1.10 (>3.88)	AST: 158 (22–71); ALT: 39 (7–50)	510 (60–305)	<10.0 (0.0–100.0)	8.1 (≤20)
20	0.88	0.77	**c.-32-13T>G/c.-32-13T>G**	3.50 (>3.88)	AST: 63 (20–64); ALT: 34 (12–42)	204 (60–305)	65.4 (0.0–100.0)	-
21	1.77	0.84	**c.-32-13T>G/c.-32-13T>G**	-	AST: 71 (22–71); ALT: 60 (7–50)	155 (60–305)	-	12.6 (≤20)
22	1.07	0.92	**c.-32-13T>G/c.-32-13T>G**	1.60 (3.88)	AST: 92 (22–71); ALT: 46 (7–50)	265 (60–305)	81.5 (0.0–100.0)	5.9 (≤20)
23	1.12	0.62	**c.2238G>C;** *c.2065G>A* **/c.2238G>C;** *c.2065G>A*	2.5 (3.88)	AST: 73 (22–71); ALT: 28 (7–50)	48 (60–305)	102.3 (0.0–100.0)	-
24		0.43	**c.1856G>A/c.2238G>C**			197 (60–305)		
25		0.95	**c.2238G>C/c.2238G>C**					
26		1.01	**c.2238G>C/c.2238G>C**					
27	1.1	0.42	**c.1441T>C/c.2238G>C/** *c.2065G>A*	2.40 (>3.88)	AST: 32 (22–71); ALT: 29 (7–50)	108 (60–305)	35.7 (0.0–100.0)	3.3 (≤20)
28		1.06	**c.-32-13T>G/c.2238G>C/** *c.2065G>A*					
29		0.48	**c.-32-13T>G/c.1547G>A**					
30		1.07	**c.32-13T>G/c.32-13T>G**					
31		0.94	**c.32-13T>G/c.2238G>C /** *c.2065G>A*					

Pathogenic or likely pathogenic variants are bolded; Pseudodeficiency mutations are made red and italicized. Transcript number: NM_000152.3; Genome build: GRCh37.

**Table 3 IJNS-06-00089-t003:** Biochemical Parameters of Screened Patients defined as Suspected LOPD.

	Newborn Screening			Confirmatory Testing				
Suspected LOPD Patients	GAA#1 (>2.10 micromole/L/h)	GAA#2 (>2.10 micromole/L/h)	Genotype	GAA	AST and ALT (normal range for lab, U/L)	CK (normal range for lab, U/L)	BNP (normal range for lab, pg/mL)	Hex4 (normal range for lab, (mmol/mol creatinine)
1	0.91	0.2	**c.-32-13T>G/**c.692+3G>C	2.1	AST: 30 (24–72); ALT: 30 (17–63)	150	ND	3.3
2	2.06	1.99	**c.-32-13T>G/**c.1594G>A	3.8	AST: 41 (24–72); ALT: 30 (17–63)	134	ND	3.0
3	1.48	0.67	**c.-32-13T>G/**c.546G>A	1.6	AST: 42 (24–72); ALT: 37 (17–63)	ND	ND	6.0
4	1.85	0.98	**c.-32-13T>G/**c.266G>A;c.1377C>G	1.4	AST: 39 (24–72); ALT: 32 (17–63)	293	ND	8.3
5	1.04	0.53	**c.-32-13T>G/**c.2003 A>G	0.6 (>3.88)		107 (26-192)		2.4 (<20)
6	0.61	0.41	**c.-32-13T>G/**c.1721T>C	0.8	AST: 96 (24–72); ALT: 84 (17–63)	377	ND	4.9
7	0.52	0.45	**c.-32-13T>G/**c.1291_1299del9	1.9	AST: 83 (20–70); ALT: 48 (17–63)	486	ND	9.4
8	0.56	0.63	**c.-32-13T>G/**1655T>C	2.5	AST: 156 (24–72); ALT: 103 (17–63)	668	ND	6.0
9	ND	1.34	**c.-32-13T>G/**c.533G>A	3.7	AST: 25 (24–72); ALT: 19 (17–63)	86	ND	2.5
10	1.3	2.01	**c.-32-13T>G/** c.862G>A;*c.271G>A*	2.0	AST: 32 (24–72); ALT: 31 (17–63)	406	ND	3.4
11	1.58	0.64	**c.-32-13T>G/** c.841C>T	1.4	AST: 23 (24–72); ALT: 42 (17–63)	136	ND	6.4
12	0.58 umol/L/h	1.84 umol/L/h	**c.1A>G/**c.1345C>T	1.90 (>3.88)	AST: 44 (15–41 U/L) ALT: 26 (12–42)	241 (15-200 U/L)	-	8.4 (≤20 mmol/mol creatinine)
13	1.02 umol/L/h	1.22 umol/L/h	**c.2560C>T/**c.1888+5G>T;*c.2065G>A*	3.80 (3.88)	AST: 112 (20–64); ALT: 29 (12–42)	69 (60–305)	407.3 (0.0–100.0)	4.6 (≤20 mmol/mol creatinine)
14	<0.19 umol/L/h	0.12 umol/L/h	**c.2236T>C/**c.700A>G/C	0.90 (>3.88)	AST: 52 (22–71); ALT: 36 (7–50)	142 (60–305)	44.2 (0.0–100.0)	15.7 (≤20)
15	1.94 umol/L/h	2.06 umol/L/h	**c.1478C>T/**1194+3G>C	6.05 (>3.88)	-	86 (60–305)	-	<4.4 (≤20)
16	1.97 umol/L/h	1.16 umol/L/h	**c.784G>A/c.859-19G>A/**c.1392G>C	4.00 (>3.88)	AST: 92 (20–64); ALT: 42 (12–42)	201 (60–305)	55.3 (0.0–100.0)	6.1 (≤20)
17	0.98 umol/L/h	0.58 umol/L/h	**c.2105G>T/**c.1124G>A	1.20 {>3.88)	AST: 64 (22–71); ALT: 20 (7–50)	85 (60–305)	-	3.0 (≤20)
18	1.59 umol/L/h	0.95 umol/L/h	**c.-32-13T>G/**c.692+3G>C	2.70 (>3.88}	AST:73 (22–71); ALT:19 (7–50)	65 (60–305)	52.7 (0.0–100.0)	3.3 (≤20)
19	1.19	1.27	**c.1655T>C/**c.1888+5G>T/*c.2065G>A*	0.43 (1.29-25.7)	AST: 60 (22–71); ALT: 25 (7–50)	110 (60–305)	19.3 (0.0–100.0)	-
20	-	1.27	**c.-32-13T>G/**c.705G>A/*c.(1726G>A;c.2065G>A)*	1.80 (3.88)	AST 80 (22–71); ALT: 41 (7–50)	86 (60–305)	30.0 (0.0–100.0)	3.3 (≤20)
21	1.64	1.1	**c.-32-13T>G/**c.650C>T	4.10 (>3.88)	AST: 65 (22–71); ALT: 35 (7–50)	68 (60–305)	33.8 (0.0–100.0)	5.7 (≤20)
22	1.02	0.85	**c.1655T>C/**c.664G>A	2.01 (>3.88)	AST:69 (20–64); ALT: 41 (12-42)	94 (60–305)	57.4 (0.0–100.0)	6.2 (≤20)
23	-	0.42	**c.-32-13T>G/**c.1103G>A	-	AST: 126 (20–64); ALT:85 (12–42)	407 {60–305)	-	7.2 (≤20)
24	-	1.08	**c.258dupC/**c.1909C>A/*c.(1726G>A;c.2065G>A)*	-	AST: 64 (22–71); ALT: 38 (7–50)	102 (60–305)	13.0 (0.0–100.0)	3.4 (≤20)
25	1.98	1.17	**c.1552-3C>G/**c.1378G>A	-	AST: 91 (22–71); ALT: 46 (7–50)	123 (60–305)	29.1 (0.0–100.0)	3.7 (≤20)
26	-	1.49	**c.1655T>C/**c.266G>A	2.10 (>3.88)	AST: 56 (22–71); ALT: 36 (7–50)	88 3 60–305)	-	3.8 (≤20)
27		0.42	**c.-32-13T>G/**c.2467A>T					
28		1.63	**c.1504A>G/**c.2467A>T					
29		1.36	**c.2560C>T/**c.726G>A					
30		0.93	**c.2238G>C/**c.2467A>T					

Transcript number: NM_000152.3; Genome build: GRCh37. Pathogenic or likely pathogenic variants are bolded; VUSs are normal; Pseudodeficiency mutations are made red and italicized; benign or likely benign are bolded and in blue. Variants were checked in the Pompe Disease Mutation Database, available at http://www.pompevariantdatabase.nl or https://www.ncbi.nlm.nih.gov/clinvar.

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
