# Peer review of "Newborn Screening for Pompe Disease: Pennsylvania Experience"

_2409-515X, 2020, doi:10.3390/ijns6040089_

Round 1
Reviewer 1 Report
This is a very nice paper describing the experience with newborn screening for Pompe disease in the state of Pennsylvania. I have several comments but all are relatively minor.
- What do the open circles represent on the figures? Unless I missed something, these are not explained on the figures or in the legend.
- Screening has been ongoing now in PA for 3.5 years so at least some of the patients have 2-3 years of followup. Have any of the LOPD or suspected LOPD patients developed symptoms over time and if so, are any of them now on treatment? A comment should be included regarding this.
- The first sentence of the conclusion states that the incidence of Pompe disease has increased since newborn screening began. Of course, this is not the case. Newborn screening has taught us that previous estimates of incidence were too low and many late onset cases may have been undiagnosed previously. Newborn screening also suggests that the fraction of cases represented by IOPD is less than previously estimated.
Author Response
This is a very nice paper describing the experience with newborn screening for Pompe disease in the state of Pennsylvania. I have several comments but all are relatively minor.
- What do the open circles represent on the figures? Unless I missed something, these are not explained on the figures or in the legend.
They are outliers. We added the meaning of open circles to the legend.
- Screening has been ongoing now in PA for 3.5 years so at least some of the patients have 2-3 years of follow up. Have any of the LOPD or suspected LOPD patients developed symptoms over time and if so, are any of them now on treatment? A comment should be included regarding this.
This paper focuses on the newborns who had abnormal NBS for Pompe. We are working on another paper to report the outcomes of patients with IOPD and LOPD detected through NBS very soon. In the second paper, we will discuss our follow up tools and outcomes of patients in detail. We added short-term outcomes of IOPD cases.
- The first sentence of the conclusion states that the incidence of Pompe disease has increased since newborn screening began. Of course, this is not the case. Newborn screening has taught us that previous estimates of incidence were too low and many late onset cases may have been undiagnosed previously. Newborn screening also suggests that the fraction of cases represented by IOPD is less than previously estimated.
We changed the conclusion and also revised the table 1 and reported the incidence of IOPD, LOPD and all forms in PA.
Reviewer 2 Report
This study adds to the growing body of literatures on the epidemiology and post-newborn screening clinical results of Pompe disease. Overall, it is a well-constructed report with abundant useful information for both public health professionals and clinicians. Some minor details are worth fixing:
- Page 2, line 52, there should be a new version of the referenced book, Reference 1.
- Page 2, line 55, it’s helpful to point out the root cause of the disorder is the pathological mutations on GAA allele.
- Page 2, line 57, there are more up-to-date references, such as reference 21 and 22.
- Page 2, line 97, reference 20 may not be the most relevant one about Pompe’s addition on RUSP.
- Page 3, line 108, the 18% cutoff value is based on only ~ 1k sample and 9 confirmed cases. Have you re-evaluated the cutoff after more data are available?
- Page 3, Figure 1, footnote 1, “within 48 hours” of what?
- Page 3, line 130, “cut off” and “cutoff” both have been used through out the article. I would recommend just use cutoff to be consistent.
- Page 4, line 135, <2.1 or < 2.10, please be consistent.
- Page 4, line 139, a brief description of the referral would be more natural here. For example, who refer the patients, how many metabolic centers, etc.
- Page 4, line 145, AST, ALT, CH, Hex4 are all familiar terms to readers with good knowledge of Pompe disease, but spelling them out initially would help other readers.
- Page 4, line 146, “This practice has changed” seems redundant.
- Page 4, Table 1, an interesting and unconventional lay out. You might consider a slight different layout with more information such as:
|
|
N |
Incidence |
|
Pompe |
|
|
|
IOPD |
2 |
1:xxxxxx |
|
LOPD |
31 |
1:xxxxxx |
|
Suspected LOPD |
30 |
1:xxxxxx |
|
IOPD+LOPD |
|
|
|
IOPD+LOPD+suspected LOPD |
|
|
|
Carriers |
35 |
|
|
Pseudo |
15 |
|
- Page 5, Table 2 and Table 3, please enlarge the font
- Page 7, line 221, “Questions raised”, there is no discussion about IOPD. If the reason being the sample size is too small to have statistical inference, please add an explanation for readers.
- Page 9, line 283, “Increased incidence is largely due to…” the statement might be true, but some other factors discussed later might also play a large role. For example, California recently reported an IOPD+LOPD+Suspected IOPD ~ 1:25,000 with similar methodologies but no repeat GAA screening. It should result in higher false positives as you discussed later about Illinois and Missouri, but not necessarily significantly lower prevalence.
- Page 10, line 330, the first sentence feels abrupt.
- Page 10, line 344, “… increased after initiation of NBS” is easy to cause misunderstanding, as if NBS is the reason for higher Pompe incidence. First, the Pompe disease incidence is always there, just the detected or diagnosed Pompe cases have increased; second, the increase is comparing what? Need more clarification.
Author Response
This study adds to the growing body of literatures on the epidemiology and post-newborn screening clinical results of Pompe disease. Overall, it is a well-constructed report with abundant useful information for both public health professionals and clinicians. Some minor details are worth fixing:
- Page 2, line 52, there should be a new version of the referenced book, Reference 1.
The 8e is the last that we know of.
- Page 2, line 55, it’s helpful to point out the root cause of the disorder is the pathological mutations on GAA allele.
We edited the first sentence as suggested.
- Page 2, line 57, there are more up-to-date references, such as reference 21 and 22.
Pathologic changes in muscle usually begin long before patients present with symptoms (2). Reference #2 is the most relevant.
- Page 2, line 97, reference 20 may not be the most relevant one about Pompe’s addition on RUSP.
We added a new reference.
- Page 3, line 108, the 18% cutoff value is based on only ~ 1k sample and 9 confirmed cases. Have you re-evaluated the cutoff after more data are available?
Yes the cutoffs are reviewed annually.
- Page 3, Figure 1, footnote 1, “within 48 hours” of what?
We added -after receiving the first abnormal screening result- .
- Page 3, line 130, “cut off” and “cutoff” both have been used through out the article. I would recommend just use cutoff to be consistent.
Cutoff is used as suggested
- Page 4, line 135, <2.1 or < 2.10, please be consistent.
Corrected in the text and tables.
- Page 4, line 139, a brief description of the referral would be more natural here. For example, who refer the patients, how many metabolic centers, etc.
It was written that there are 7 referral centers . We added by pediatricians and department of health newborn screening program.
- Page 4, line 145, AST, ALT, CH, Hex4 are all familiar terms to readers with good knowledge of Pompe disease, but spelling them out initially would help other readers.
We spelled each in the text.
- Page 4, line 146, “This practice has changed” seems redundant.
The sentence is deleted as suggested.
- Page 4, Table 1, an interesting and unconventional lay out. You might consider a slight different layout with more information such as:
|
|
N |
Incidence |
|
Pompe |
|
|
|
IOPD |
2 |
1:xxxxxx |
|
LOPD |
31 |
1:xxxxxx |
|
Suspected LOPD |
30 |
1:xxxxxx |
|
IOPD+LOPD |
|
|
|
IOPD+LOPD+suspected LOPD |
|
|
|
Carriers |
35 |
|
|
Pseudo |
15 |
|
We changed the table as suggested.
- Page 5, Table 2 and Table 3, please enlarge the font
The font is enlarged .
- Page 7, line 221, “Questions raised”, there is no discussion about IOPD. If the reason being the sample size is too small to have statistical inference, please add an explanation for readers.
We added the limitations due to small sample size and discussed their values as well.
- Page 9, line 283, “Increased incidence is largely due to…” the statement might be true, but some other factors discussed later might also play a large role. For example, California recently reported an IOPD+LOPD+Suspected IOPD ~ 1:25,000 with similar methodologies but no repeat GAA screening. It should result in higher false positives as you discussed later about Illinois and Missouri, but not necessarily significantly lower prevalence.
We do not say that second GAA screening increase the detection of LOPD or suspected LOPD.
Since there are more false positives reported by MO or IL , measuring GAA second time might decrease false positives.
- Page 10, line 330, the first sentence feels abrupt.
We think that the following sentence explains the challenge.
- Page 10, line 344, “… increased after initiation of NBS” is easy to cause misunderstanding, as if NBS is the reason for higher Pompe incidence. First, the Pompe disease incidence is always there, just the detected or diagnosed Pompe cases have increased; second, the increase is comparing what? Need more clarification.
The first sentence of the conclusion is rephrased.
Reviewer 3 Report
1. In the Materials and Methods section it is stated that the "resulting products are combined...", but it appears that one disc is used for all enzyme assays. Please clarify what was combined.
2. Are the authors aware of any false negative screens (i.e., infants with Pompe Disease that were not identified by newborn screening)?
3. The authors have several "Questions raised by Screening Results", but don't give definitive answers. Although they were able to detect significant differences in some median values, they should more clearly state that the biochemical results don't allow the various types of Pompe Disease to be identified. The one determinant biochemical result that is not mentioned is that Hex4 appears to distinguish IOPD from LOPD.
4. In the Introduction the authors state that about 1/3 of patients with Pompe Disease have IOPD. That is not born out by their data, or the data from IL or MO that they cite. This is worth a comment. PA also seems to have a slightly lower incidence of IOPD that IL or MO.
5. The PA model is for two positive DBS screens before DNA sequencing is done. This could cause a delay in diagnosis if the second specimen is not received in a timely manner. What do they do if the second specimen is not received?
6. The authors state in the Discussion that the two IOPD patients were successfully treated. How is success defined?
Author Response
1.In the Materials and Methods section it is stated that the "resulting products are combined...", but it appears that one disc is used for all enzyme assays. Please clarify what was combined.
The sentence should be changed to read as follows: “The enzymatic reactions are quenched with 50 % ethyl acetate: methanol followed by a liquid-liquid extraction using 50% ethyl acetate: water.”
- Are the authors aware of any false negative screens (i.e., infants with Pompe Disease that were not identified by newborn screening)?
We are not aware of any cases missed by PA NBS.
- The authors have several "Questions raised by Screening Results", but don't give definitive answers. Although they were able to detect significant differences in some median values, they should more clearly state that the biochemical results don't allow the various types of Pompe Disease to be identified. The one determinant biochemical result that is not mentioned is that Hex4 appears to distinguish IOPD from LOPD.
We gave more definitive answers in the result and discussion..
- In the Introduction the authors state that about 1/3 of patients with Pompe Disease have IOPD. That is not born out by their data, or the data from IL or MO that they cite. This is worth a comment. PA also seems to have a slightly lower incidence of IOPD that IL or MO.
We edited the lines 77-87 on page 3. We also commented on lower incidence of IOPD in PA in the text and the table 1.
- The PA model is for two positive DBS screens before DNA sequencing is done. This could cause a delay in diagnosis if the second specimen is not received in a timely manner. What do they do if the second specimen is not received?
There is no incident that the second sample was not done. Delay in obtaining the second sample may risk the early detection of IOPD. We commented on that in the discussion. The repeat specimen is requested within 48 hrs. In addition, it is recommended on the report to discuss the case with one of the metabolism referral centers if newborn has cardiomyopathy or symptoms of Pompe disease such as hypotonia and feeding issues. A second specimen is necessary to complete the sequencing test due to depletion of blood on the initial specimen on which all other newborn screening tests are performed.
The authors state in the Discussion that the two IOPD patients were successfully treated. How is success defined?
We chose to delete “ successfully” , and wrote more about outcomes of each IOPD case.
Reviewer 4 Report
Summary
Focus of this paper is on utility of biochemical GAA values, other biochemical markers [muscle damage (CK), liver cell damage (AST/ALT) or urine Hex4 marker] and DNA sequencing data for evaluating newborns for Pompe disease. While the GAA values on their own do not show much difference within disease sub-type there may be value when these markers are used together in machine learning algorithms. This experience is therefore critical to public health screening initiatives. Some observations are not discussed such as IOPD (most likely due to small N). Also missing from discussion is the followup data for the IOPD cases, although it is stated they were treated successfully. However, that success is not described. Biochemical markers and/or genotyping could be utilized as part of NBS either to quickly assess if a case is IOPD or LOPD or non-disease. There is a sentence and reference in the discussion, but it is vague. The disease subtyping is critical for those 2 individuals who were IOPD as treatment approaches are diffrerent between IOPD and LOPD and dependent on CRIM status. The probability of having a low GAA and elevated CK, ALT or Hex4 is a strong indicator of IOPD. Although CK levels is often thought by clinicians and experts to be a non-specific marker of muscle damage (given elevations due to birth trauma), this level rapidly reduces after delivery. Further, in combination with other markers it may be a better classifier in not only sorting FP but also disease sub-typing. The issue of FP per se, is important, but also important is that IOPD cases be not delayed as they may have permanent damage if therapy is not initiated immediately.Thus whether emphasis is placed on sequencing or biochemical markers, some discussion as to what is the rationale and future directions should be beneficial to the readers.
More detailed notes:
Line 43: the + sign spacing and starting sentence with a number should be corrected.
Line 54: Muscle damage remark is not connected to the biomarker CK. The author should provide some rationale behind the markers and their importance (cite Kishnani et al., 2006 or Burton et al., 2020in IJNS).
Line 57: 1 in 40,000 needs reference and geography or ethnicity.
Line 68: (minor) should say 18 years. There are minor inconsistencies like this that should be corrected.
Line 73: (minor) “These factors importantly ….disease”. Does this sentence add value given prior sentence?
Line 75: (minor) spacing before reference ‘(9)’.
Line 79: Use CRIM- consistently. The use of the word ‘may’ in context of immune response is confusing and suggests that in CRIM- patients sometimes the immune system may not mount a response. Is that the right message? Does that mean that sometimes immunomodulation treatment with methotrexate is not required?
Line 83-86: This introductory statement is very helpful but needs to be reworked and connected. The previous paragraph suggests CRIM- , and need for immune modulation prior to ERT, but the Taiwanese experiences are not addressing that point well as it seems the reference is being used for stating they initiated NBS program. It would help if the issues of diagnostic odyssey, introduction of Pompe screening and CRIM -/treatment decisions paragraphs are dealt with individually so the readers are able to follow.
Line 84: consider reworking this statement. Provide more specificity as otherwise it is too broad and not informative.
Line 92: Is ‘increased storage of glycogen’ the right message? Does that create dysfunction? LSD experts understand the process but a general reader will not appreciate this. This sentence needs improvement.
Line 98: It would help if the point was further elaborated that State public health laboratories (PHL) do not always wait for screening mandated by RUSP and that some states like PA and others have introduced pilots and screening for disorders before it was RUSP approved. This goes to show the importance of the independent process and value of PHL screening, and the way it is organized in the USA. As this allows multiple approaches to introduction of a new screening disorder or approach or concept in NBS.
Line 100-102: (minor) ‘Our aim is three-fold: to augment state-based knowledge of the disease and its diagnosis; to document the benefits of NBS for Pompe patients in Pennsylvania; and to encourage the speedy adoption of NBS for Pompe by other states and countries’. Is this is a PA PHL Aim or Aim of the paper?
Line 104: PerkinElmer. Should this have a trade mark or city state? Does PE do the GAA sequencing test? Who did the other tests? Are CK levels from serum or on 3mm dried blood spot discs?
Line 107: The second specimen request is a process that may cause delay. This could harm the IOPD cases as described in the Chien et al references, especially if not performed within days. Given only 2 IOPD cases, this will not be easy but the overall algorithm may not be able to help the most vulnerable cases (IOPD), who needs immunomodulation therapy and for which CRIm status and sequence data may be the only way to quickly provide information. However, late onset disease may not have such a stringent requirement. The risks LOPD may be loss to followup or disparity.
Line 122: a reference for the MS/MS method should be used.
Line 129: the equation and the acronymns should be better described and connected in the paragraph (Lines 125-120).
Line 133: Is GAA gene sequencing done by the NBS program, metabolic center/clinic or PE? Is it Sanger or NGS? What does 7-10 days from reflex mean? When is that send-out done? If sequencing is done after the second GAA levels are deermined and since that involves recollection, significant time would elapse by the time the results come back. In case of IOPD that means the babies are not getting referred and babies are not on ERT. The authors should discuss this experience in the discussion section of f/u on the two IOPD cases or provide some details in the methods section. That the IOPD cases were treated within 3 weeks or 10th day of life is important. However, the infants were already in the hospital. The risk is that this is still not as timely as one would expect based on consensus and where days of life matter.Cases lost to follow-up is another consideration of providing definitive info as part of NBS. Any evidence of health of IOPD patients beyond 1 yr will also go to prove outcome and how the NBS algorithm is working. This would probably provide data to suggest if the algorithm makes a difference for IOPD cases.
Line 134: The GAA biochem TAT is <72 hrs but the sequencing is 7-10 days. Thus again with the serial approach of second GAA collection that will add days for collection and 7-10 days for return of DNA data, the value of sequencing is only confirmatory and too late; the outcomes on IOPD patients could be a challenge. In line 182 it is clear that if CK alongside GAA values is used it can help in sorting out sequencing cases on the first tier biochem itself. Yet the treatment did not start till 3 weeks while the other IOPD case was on 10 days of life. One wonders what the outcomes were given earlier experience where IOPD cases when not treated within 10 days was not so great. Although the IOPD cases are small in number (2 in PA and 3 in IL based on the recent studies) this may add up to several humdred babies across the US.
Line 162 and 164: consider using Figure instead of figure. Consider using the word Tables.
Line 185 The infant was treated as CRIM- or CRIM+ but Table 2 does not show predicted CRIM status wich is needed to initiate ERT and possible from DNA data. 90% of CRIM data is predictable prom DNA either because they already exist and breed true or is predcted due to protein truncations. Outcome data on these infants are also missing or unavailable but these data may help understand what is working and what still is a challenge. The text mentions that the variants were previously described but source was not provided (e.g.Erasmus). PA NBS may be only taking the responsibility of just screening the GAA values and not concerned with the other marker values or DNA data. The protocol and priorities should therefore be described.
The two IOPD cases based on Erasmus DB
c.759del/c.1551+1G>C CRIM prediction -/?
- 525del/c.1694-1697delTCTC CRIM Prediction -/-
Line 213 Pseudodeficiency Carriers and Carriers: Some discussion should be devoted to what families valued. It is possible that if those families have additional children they may not be alarmed by low GAA values if inherited, on the other hand the alleles may continue to create anxiety for parents and perhaps GC consults can help avoid anxiety and odyssey and educate the parents (evident in Table 2 and 3). This may be important information for other NBS programs.
The value of DNA data may be also valuable for subsequent pregnancies as they are high-risk and in those families.
Line 224: Does NBS GAA levels distinguish different types of GAA?
There are various factors that goes into the GAA measurement at the biochemical level. There is seasonal variation, variation due to punch size. At the molecular level pseudodeficiencies may be present in combination with pathogenic (disease causing) variants. So a single biochemical value associated with disease sub-type will be a difficult classifier. Machine learning algorithms that use multiple factors to accurately classify disease may be possible like CLIR algorithm, but these are not always intuitive.
The discussion is centered more around disease vs non-disease states (carriers and pseudodeficiency). So the question has to be altered to reflect this as carrier is not a disease state.
In Figure 2, IOPD has no legends and the confirmatory GAA levels are higher, this is not explained well in the Table, Figure or the manuscript.
Line 232: Figure 2 should have values of individual cases rather than box-plots. Also the confirmatory values should be separated from the two NBS values as this will allow to change the scale and get better feel for the data.
Line 252: (minor point) If CK values are in the Tables why is it repeated in the Figures? The ranges are confusing. Table 3 has repeated enzyme activity values (micromole/L/hr…
Line 256: In general, patients with IOPD almost invariably have elevated urinary Hex4 and CK. This pattern is similar to what Burton et al., (2020) reports. Although the number of cases (n=2) for IOPD is low in PA to draw statistical significance this observation could be meaningful to follow and could be a classifier that can contribute to better disease classification.
Line 293: The value of molecular diagnosis is not limited to just the first visit to confirm the disease but could also be in decision of treatment or followup tests for NBS team and clinicians as they consider to amend algorithms or arrive at disease classification and treatment approach.
Line 295-298: The disparity avoided is important and the authors should make these sentences more clear. What leads to uncertainity and how is it addressed? How is the genetic test paid for and what happens there… is it rejected or it takes months or what occurs? When genetic tesing is included as part of NBS or f/u clinic routine testing irrespective of insurance carried by family that may reduce disparity. Another question is how is it paid for. Educating other groups about this issue could be helpful.
Line 302: CK may be also be able to suggest IOPD cases as the values are elevated. The marker can be evaluated as part of NBS after the first GAA levels are determined. Hex4 can only be done from urine and ALT from larger blood volumes, and while important for LOPD assessment not that helpful for IOPD. The authors should consider this point given statement on all genotypes in line 321.
Line 336: The definition of a false positive in PA vs. IL vs. MO is not obvious to the reader. How are they determined? It may be important to discuss the differences in algorithms in context of patient benefits, and not just FP rates or referral burden. In both the MO and the PA protocols the IOPD cases would not be detected outright although the LOPD cases may be better picked up with lower referral burden by the PA algorithm. MO (Klug et al., 2020 IJNS) never considered genetic testing initially, unlike IL (Burton et al., 2020 IJNS) or PA. It may also be the FP rates is a function of test methodology.
Author Response
Focus of this paper is on utility of biochemical GAA values, other biochemical markers [muscle damage (CK), liver cell damage (AST/ALT) or urine Hex4 marker] and DNA sequencing data for evaluating newborns for Pompe disease. While the GAA values on their own do not show much difference within disease sub-type there may be value when these markers are used together in machine learning algorithms. This experience is therefore critical to public health screening initiatives. Some observations are not discussed such as IOPD (most likely due to small N). Also missing from discussion is the followup data for the IOPD cases, although it is stated they were treated successfully. However, that success is not described. Biochemical markers and/or genotyping could be utilized as part of NBS either to quickly assess if a case is IOPD or LOPD or non-disease. There is a sentence and reference in the discussion, but it is vague. The disease subtyping is critical for those 2 individuals who were IOPD as treatment approaches are diffrerent between IOPD and LOPD and dependent on CRIM status. The probability of having a low GAA and elevated CK, ALT or Hex4 is a strong indicator of IOPD. Although CK levels is often thought by clinicians and experts to be a non-specific marker of muscle damage (given elevations due to birth trauma), this level rapidly reduces after delivery. Further, in combination with other markers it may be a better classifier in not only sorting FP but also disease sub-typing. The issue of FP per se, is important, but also important is that IOPD cases be not delayed as they may have permanent damage if therapy is not initiated immediately.Thus whether emphasis is placed on sequencing or biochemical markers, some discussion as to what is the rationale and future directions should be beneficial to the readers.
We thank reviewer#4 for thoughtful and comprehensive suggestions and comments. Our sample size is not large enough to be able to implement any machine learning approach. In this paper, our main aim is to present cases detected through Pompe NBS in PA , and discuss their biochemical values of NBS and initial visit. We are planning to do more statistical methods to build predictive model when we have larger sample. We also do not present any short term outcomes in this paper since we have a follow up paper that will discuss diagnosis, follow up , treatment decision , and outcomes.
More detailed notes:
Line 43: the + sign spacing and starting sentence with a number should be corrected.
It is corrected as suggested.
Line 54: Muscle damage remark is not connected to the biomarker CK. The author should provide some rationale behind the markers and their importance (cite Kishnani et al., 2006 or Burton et al., 2020in IJNS).
The introduction is edited to accommodate the reviewer’s suggestions .
Line 57: 1 in 40,000 needs reference and geography or ethnicity.
We added the references .
Ausems MG, Verbiest J, Hermans MP, Kroos MA, Beemer FA, Wokke JH, Sandkuijl LA, Reuser AJ, van der Ploeg AT. Frequency of glycogen storage disease type II in The Netherlands: implications for diagnosis and genetic counselling. Eur J Hum Genet. 1999, 7:713–6.
Martiniuk F, Chen A, Mack A, Arvanitopoulos E, Chen Y, Rom WN, Codd WJ, Hanna B, Alcabes P, Raben N, Plotz P. Carrier frequency for glycogen storage disease type II in New York and estimates of affected individuals born with the disease. Am J Med Genet. 1998;79:69–72
Line 68: (minor) should say 18 years. There are minor inconsistencies like this that should be corrected.
Corrected
Line 73: (minor) “These factors importantly ….disease”. Does this sentence add value given prior sentence?
We changed the first paragraph based on another reviewer’s suggestion.
Line 75: (minor) spacing before reference ‘(9)’.
Corrected.
Line 79: Use CRIM- consistently. The use of the word ‘may’ in context of immune response is confusing and suggests that in CRIM- patients sometimes the immune system may not mount a response. Is that the right message? Does that mean that sometimes immunomodulation treatment with methotrexate is not required?
We deleted “may“
Line 83-86: This introductory statement is very helpful but needs to be reworked and connected. The previous paragraph suggests CRIM- , and need for immune modulation prior to ERT, but the Taiwanese experiences are not addressing that point well as it seems the reference is being used for stating they initiated NBS program. It would help if the issues of diagnostic odyssey, introduction of Pompe screening and CRIM -/treatment decisions paragraphs are dealt with individually so the readers are able to follow.
We discuss the treatment and CRIM and we believe they are connected. We edited the introduction to make it more connected.. We discuss the basics of Pompe, diagnostic odyssey, basics of treatment, and importance of early diagnosis and newborn screening.
Line 84: consider reworking this statement. Provide more specificity as otherwise it is too broad and not informative.
We respectfully disagree and think this paragraph summarizes importance of NBS for Pompe
Line 92: Is ‘increased storage of glycogen’ the right message? Does that create dysfunction? LSD experts understand the process but a general reader will not appreciate this. This sentence needs improvement.
This is the finding reported by Chien YH et al. We do not think more detailed info is need for this journal readers.
Line 98: It would help if the point was further elaborated that State public health laboratories (PHL) do not always wait for screening mandated by RUSP and that some states like PA and others have introduced pilots and screening for disorders before it was RUSP approved. This goes to show the importance of the independent process and value of PHL screening, and the way it is organized in the USA. As this allows multiple approaches to introduction of a new screening disorder or approach or concept in NBS.
PA started NBS for POMPE one year after RUSP. There was no a pilot in PA.
It is very important to publish state experience on NBS for Pompe. The existence of differing approaches according to protocols of different states may be helpful until a clear agreed upon algorithm emerges.
The existence of at least three different state-level approaches to NBS for Pompe (PA, MO, IL) makes it possible to assess the effectiveness of each and move toward achieving a national consensus about best protocols.
Line 100-102: (minor) ‘Our aim is three-fold: to augment state-based knowledge of the disease and its diagnosis; to document the benefits of NBS for Pompe patients in Pennsylvania; and to encourage the speedy adoption of NBS for Pompe by other states and countries’. Is this is a PA PHL Aim or Aim of the paper?
This is the aim of the paper to show the value of NBS and share our experience.
Line 104: PerkinElmer. Should this have a trade mark or city state? Does PE do the GAA sequencing test? Who did the other tests? Are CK levels from serum or on 3mm dried blood spot discs?
Newborn screening tests ( GAA enzyme and GAA full gene sequencing are done by PerkinElmer lab as written in the material and methods. PerkinElmer is in Pittsburgh, PA and it is listed on Page 1, line 22. The confirmatory tests GAA, AST, ALT, CK, Hex4 are done by the metabolic centers. CK levels are measured in serum. We edited the test to make this more clear.
Line 107: The second specimen request is a process that may cause delay. This could harm the IOPD cases as described in the Chien et al references, especially if not performed within days. Given only 2 IOPD cases, this will not be easy but the overall algorithm may not be able to help the most vulnerable cases (IOPD), who needs immunomodulation therapy and for which CRIm status and sequence data may be the only way to quickly provide information. However, late onset disease may not have such a stringent requirement. The risks LOPD may be loss to followup or disparity.
We added a sentence and discuss the potential delay in detecting IOPD when two tests are done .
Line 122: a reference for the MS/MS method should be used.
It is added in the reference.
Line 129: the equation and the acronymns should be better described and connected in the paragraph (Lines 125-120).
We believe we have adequately described the equation, with commonly used abbreviations, in the body of the text as it reads below:
Enzyme activities (in units of µmol/L/h) are defined as the amount of substrate hydrolyzed by the enzyme in the reaction and are determined by calculating the ion abundance ratio of product to internal standard, multiplied by the volume and concentration of internal standard, divided by the response factor ratio of product to internal standard, sample incubation time, and sample blood volume (Enzyme activity = (P/IS)*[IS]*VIS/RF/3.1 µL/time). A sample volume of 3.1 µL is assigned to a 1/8 inch DBS punch.
Line 133: Is GAA gene sequencing done by the NBS program, metabolic center/clinic or PE? Is it Sanger or NGS? What does 7-10 days from reflex mean? When is that send-out done? If sequencing is done after the second GAA levels are deermined and since that involves recollection, significant time would elapse by the time the results come back. In case of IOPD that means the babies are not getting referred and babies are not on ERT. The authors should discuss this experience in the discussion section of f/u on the two IOPD cases or provide some details in the methods section. That the IOPD cases were treated within 3 weeks or 10th day of life is important. However, the infants were already in the hospital. The risk is that this is still not as timely as one would expect based on consensus and where days of life matter.Cases lost to follow-up is another consideration of providing definitive info as part of NBS. Any evidence of health of IOPD patients beyond 1 yr will also go to prove outcome and how the NBS algorithm is working. This would probably provide data to suggest if the algorithm makes a difference for IOPD cases.
-PerkinElmer (as part of NBS )
-NGS
-Sequencing is performed at the same location as the enzyme test (PE).
-As stated previously we discussed potential delay in detecting IOPD based on current algorithm-
Line 134: The GAA biochem TAT is <72 hrs but the sequencing is 7-10 days. Thus again with the serial approach of second GAA collection that will add days for collection and 7-10 days for return of DNA data, the value of sequencing is only confirmatory and too late; the outcomes on IOPD patients could be a challenge. In line 182 it is clear that if CK alongside GAA values is used it can help in sorting out sequencing cases on the first tier biochem itself. Yet the treatment did not start till 3 weeks while the other IOPD case was on 10 days of life. One wonders what the outcomes were given earlier experience where IOPD cases when not treated within 10 days was not so great. Although the IOPD cases are small in number (2 in PA and 3 in IL based on the recent studies) this may add up to several humdred babies across the US.
We agree that IOPD cases should be detected as early as possible.
Line 162 and 164: consider using Figure instead of figure. Consider using the word Tables.
Corrected as suggested
Line 185 The infant was treated as CRIM- or CRIM+ but Table 2 does not show predicted CRIM status wich is needed to initiate ERT and possible from DNA data. 90% of CRIM data is predictable prom DNA either because they already exist and breed true or is predcted due to protein truncations. Outcome data on these infants are also missing or unavailable but these data may help understand what is working and what still is a challenge. The text mentions that the variants were previously described but source was not provided (e.g.Erasmus). PA NBS may be only taking the responsibility of just screening the GAA values and not concerned with the other marker values or DNA data. The protocol and priorities should therefore be described.
The two IOPD cases based on Erasmus DB
c.759del/c.1551+1G>C CRIM prediction -/?
- 525del/c.1694-1697delTCTC CRIM Prediction -/-
Each cases’ CRIM status was discussed in the text. We added outcomes of IOPD cases . We use both Erasmus and ClinVar .
We added “ Variants were checked in the Pompe Disease Mutation Database, available at http://www.pompevariantdatabase.nl or https://www.ncbi.nlm.nih.gov/clinvar.”
Line 213 Pseudodeficiency Carriers and Carriers: Some discussion should be devoted to what families valued. It is possible that if those families have additional children they may not be alarmed by low GAA values if inherited, on the other hand the alleles may continue to create anxiety for parents and perhaps GC consults can help avoid anxiety and odyssey and educate the parents (evident in Table 2 and 3). This may be important information for other NBS programs.
The value of DNA data may be also valuable for subsequent pregnancies as they are high-risk and in those families.
The purpose of NBS is to detect patients with disease ( IOPD , LOPD) . As stated in the text, cases with pseudodeficiency are not referred to metabolic centers and their NBS is not reported as positive. We choose not to discuss pseudodeficiency in this paper.
Line 224: Does NBS GAA levels distinguish different types of GAA?
There are various factors that goes into the GAA measurement at the biochemical level. There is seasonal variation, variation due to punch size. At the molecular level pseudodeficiencies may be present in combination with pathogenic (disease causing) variants. So a single biochemical value associated with disease sub-type will be a difficult classifier. Machine learning algorithms that use multiple factors to accurately classify disease may be possible like CLIR algorithm, but these are not always intuitive.
The discussion is centered more around disease vs non-disease states (carriers and pseudodeficiency). So the question has to be altered to reflect this as carrier is not a disease state.
In Figure 2, IOPD has no legends and the confirmatory GAA levels are higher, this is not explained well in the Table, Figure or the manuscript.
Larger numbers are needed to develop machine learning algorithms. There are only two IOPD cases and one who responded to the treatment much better has more residual enzyme activity. This paper presents cases detected through NBS. We are working on another paper to present cases and their outcomes in greater detail.
Line 232: Figure 2 should have values of individual cases rather than box-plots. Also the confirmatory values should be separated from the two NBS values as this will allow to change the scale and get better feel for the data.
Due to the sample size, it would be messy if we plot all the actual values rather than the boxplot. Table 1 and 2 list all individual values. Tables also include reference range for both NBS and confirmatory GAA.
Line 252: (minor point) If CK values are in the Tables why is it repeated in the Figures? The ranges are confusing. Table 3 has repeated enzyme activity values (micromole/L/hr…
Tables include each indiviual CK values. The figures have median, min and max.
Line 256: In general, patients with IOPD almost invariably have elevated urinary Hex4 and CK. This pattern is similar to what Burton et al., (2020) reports. Although the number of cases (n=2) for IOPD is low in PA to draw statistical significance this observation could be meaningful to follow and could be a classifier that can contribute to better disease classification.
IOPD number are too small to make any meaningful discussion in PA.
Line 293: The value of molecular diagnosis is not limited to just the first visit to confirm the disease but could also be in decision of treatment or followup tests for NBS team and clinicians as they consider to amend algorithms or arrive at disease classification and treatment approach.
We highlight the importance of performing GAA full sequencing as part of NBS. We also added that it can help to determine CRIM status.
Line 295-298: The disparity avoided is important and the authors should make these sentences more clear. What leads to uncertainity and how is it addressed? How is the genetic test paid for and what happens there… is it rejected or it takes months or what occurs? When genetic tesing is included as part of NBS or f/u clinic routine testing irrespective of insurance carried by family that may reduce disparity. Another question is how is it paid for. Educating other groups about this issue could be helpful.
In the text , we made it clear that uncertainty of diagnosis stems from the absence of genetic testing at the time of the initial NBS. This delay of final diagnosis also results in uncertainty around follow up plan and treatment. We discuss that this delay (or “disparity”) is avoided when the genetic test is done as part of NBS and paid by NBS.
Not sure if we understand f/u routine clinic test means irrespective of insurance carried by family ? The confirmatory tests done at initial visits are paid by families and their insurance. There is no way around this in the USA.
Line 302: CK may be also be able to suggest IOPD cases as the values are elevated. The marker can be evaluated as part of NBS after the first GAA levels are determined. Hex4 can only be done from urine and ALT from larger blood volumes, and while important for LOPD assessment not that helpful for IOPD. The authors should consider this point given statement on all genotypes in line 321.
We edited and discussed the elevated CK levels in IOPD in the previous paragraph. Urine Hex4 levels are very important in IOPD patients.
Line 321 discusses the common c.-32-13T>G mutation.
Line 336: The definition of a false positive in PA vs. IL vs. MO is not obvious to the reader. How are they determined? It may be important to discuss the differences in algorithms in context of patient benefits, and not just FP rates or referral burden. In both the MO and the PA protocols the IOPD cases would not be detected outright although the LOPD cases may be better picked up with lower referral burden by the PA algorithm. MO (Klug et al., 2020 IJNS) never considered genetic testing initially, unlike IL (Burton et al., 2020 IJNS) or PA. It may also be the FP rates is a function of test methodology.
PA definition of false positive is cases who had low enzyme activity and negative genetic test referred to metabolic center and their confirmatory GAA enzyme is within normal range. When we compare data based on this definition , Il has 234 cases with normal confirmatory GAA activity , MO
Neither MO nor IL perform genotype as a second tier test as part of NBS.
Round 2
Reviewer 4 Report
The authors did a very good job of addressing their views and this makes this paper acceptable for publication.
Minor suggestions:
Line 126: should read Figure 1
Line 157: GAA sequencing does not distinguish between Sanger and NGS based approach. This may be useful for other state Public health laboratories to know.
Line 184: please delete “Copyright © 2002-2012 by”
Line 260 and 278: The answer to the question is not relevant (see below). The biochem levels (GAA) distinguish disease vs. non-disease (carrier and pseudos). So either the question or the answer has to be adjusted. Line 260: Do NBS GAA and confirmatory GAA levels make it possible to distinguish different types of Pompe disease? Line 278: Answer: In our cohort, it was possible to differentiate LOPD and suspected LOPD cases from pseudodeficiency and carriers based on GAA enzyme levels.
Line 395: The definition of a FP on PA is not obvious. The FP definition itself is confusing to reader to understand the point. A FPR may be defined as those that are false screen positives (normal + carrier + pseudo)/total screened or out of those referred? If Pseudo is not to be discussed or outside FP definition then is the 59% FP because of normal + carrier fraction?
Neither MO nor IL performs genotype as a second tier test as part of NBS is a point, but the reference made by reviewer was to the papers from MO and IL. For e.g. the reviewer already outlined that 'MO never considered genetic testing initially (Klug et al., 2020)'.The fact that IL or MO does genotype is not the point. The authors should provide a reference or a personal communication so the FP or FPR can be verified by a reader, as otherwise it cannot be understood.
The other point to understand is by recalling babies twice the FP may be reduced and that favors Late-onset but not infantile-onset Pompe.
Author Response
Reviewer 4 :
We appreciate careful review and suggestions. We edited the paper per the reviewer’s comments/suggestions.
Line 126: should read Figure 1
Corrected.
Line 157: GAA sequencing does not distinguish between Sanger and NGS based approach. This may be useful for other state Public health laboratories to know.
Added.
Line 184: please delete “Copyright © 2002-2012 by”
Deleted.
Line 260 and 278: The answer to the question is not relevant (see below). The biochem levels (GAA) distinguish disease vs. non-disease (carrier and pseudos). So either the question or the answer has to be adjusted. Line 260: Do NBS GAA and confirmatory GAA levels make it possible to distinguish different types of Pompe disease? Line 278: Answer: In our cohort, it was possible to differentiate LOPD and suspected LOPD cases from pseudodeficiency and carriers based on GAA enzyme levels.
Corrected both the question and answer.
Line 395: The definition of a FP on PA is not obvious. The FP definition itself is confusing to reader to understand the point. A FPR may be defined as those that are false screen positives (normal + carrier + pseudo)/total screened or out of those referred? If Pseudo is not to be discussed or outside FP definition then is the 59% FP because of normal + carrier fraction?
Neither MO nor IL performs genotype as a second tier test as part of NBS is a point, but the reference made by reviewer was to the papers from MO and IL. For e.g. the reviewer already outlined that 'MO never considered genetic testing initially (Klug et al., 2020)'.The fact that IL or MO does genotype is not the point. The authors should provide a reference or a personal communication so the FP or FPR can be verified by a reader, as otherwise it cannot be understood. The other point to understand is by recalling babies twice the FP may be reduced and that favors Late-onset but not infantile-onset Pompe.
FP was reported as newborns who had normal confirmatory GAA levels in IL and MO. We used their most recent papers as references. In PA , we chose the same definition and made it clear in the text. We added a sentence at the end of results section which describes who we call FP and what their data were in PA. We also added this group in our table 1.